# Soil Geochemical Properties Influencing the Diversity of Bacteria and Archaea in Soils of the Kitezh Lake Area, Antarctica

**DOI:** 10.3390/biology11121855

**Published:** 2022-12-19

**Authors:** Qinxin Li, Nengfei Wang, Wenbing Han, Botao Zhang, Jiaye Zang, Yiling Qin, Long Wang, Jie Liu, Tao Zhang

**Affiliations:** 1College of Chemistry and Chemical Engineering, Qingdao University, Qingdao 266071, China; 2School of Chemistry and Chemical Engineering, Linyi University, Linyi 276005, China; 3First Institute of Oceanography, Ministry of Natural Resources, Qingdao 266061, China; 4Department of Bioengineering, College of Marine Sciences and Biological Engineering, Qingdao University of Science & Technology, Qingdao 266042, China; 5China Pharmaceutical Culture Collection, Institute of Medicinal Biotechnology, Chinese Academy of Medical Sciences & Peking Union Medical College, Beijing 100050, China

**Keywords:** Antarctica, microbial diversity, climate change, seasonal meltwater lakes, high-throughput sequencing, weighted gene coexpression network analysis

## Abstract

**Simple Summary:**

Global warming has always been a topic of concern for people. The polar regions are an ideal research area to study the impact of global warming because they are less affected by human activities. Seasonal meltwater lakes are a typical landscape in which parts of the soil are covered with water in the summer and exposed to air in the winter. In this study, three types of soils, including BPK (always above water), INT (sometimes above water, sometimes underwater), and SED (always underwater), were selected from the Kitezh Lake area in Antarctica to investigate the relationship between soil geochemical properties and bacterial and archaeal diversity and community structure. The key geochemical factors were found through RDA (CCA), and the network diagrams were drawn by WGCNA to find the hub OTUs. The results showed that pH, phosphate, nitrite, moisture content, ammonium, nitrate, and total carbon content all played important roles in bacterial and archaeal diversity and structure at different sites. This work provides an idea for future microbial diversity analysis, which is finding hub OTUs through WGCNA. Under the aggravation of global warming, this study also makes a contribution to predicting changes in soil microbial communities in lake regions.

**Abstract:**

It is believed that polar regions are influenced by global warming more significantly, and because polar regions are less affected by human activities, they have certain reference values for future predictions. This study aimed to investigate the effects of climate warming on soil microbial communities in lake areas, taking Kitezh Lake, Antarctica as the research area. Below-peak soil, intertidal soil, and sediment were taken at the sampling sites, and we hypothesized that the diversity and composition of the bacterial and archaeal communities were different among the three sampling sites. Through 16S rDNA sequencing and analysis, bacteria and archaea with high abundance were obtained. Based on canonical correspondence analysis and redundancy analysis, pH and phosphate had a great influence on the bacterial community whereas pH and nitrite had a great influence on the archaeal community. Weighted gene coexpression network analysis was used to find the hub bacteria and archaea related to geochemical factors. The results showed that in addition to pH, phosphate, and nitrite, moisture content, ammonium, nitrate, and total carbon content also play important roles in microbial diversity and structure at different sites by changing the abundance of some key microbiota.

## 1. Introduction

Climate change has always been a hot and far-reaching issue. Antarctica is one of the most rapidly warming regions globally, where the warming rate has exceeded 0.1 °C per decade in the half-century since 1958 [1]. In the area of Antarctica, the rising temperature causes many environmental problems, such as the retreat [2] and readvance [3] of the ice sheet, the collapse of glaciers into the sea [4,5], the rise of the global sea level [6], and the intensification of glacial meltwater [7]. One of the results for meltwater is the formation of seasonal lakes, which are particularly sensitive to environmental changes and can amplify air warming [8]. As temperature rises, lakes open earlier, and soils absorb more solar energy in the summer. The heat is transfers to the water column in winter, and ice cover reduces heat loss. Both of the processes make water warming more amplified than air warming. In seasonally formed lakes, part of the soil submerges in the summer and emerges from the water in the winter. Three different types of soils form in the lake area: below-peak soil (above the water), intertidal soil (sometimes above the water and sometimes under the water), and sediment (under the water).

According to Vincent [9], Arctic microbiota was sensitive to exhibiting rapid changes under global warming and can be viewed both as sentinels and amplifiers of global change. In the Antarctic region, bacterial communities have been shown to adapt more quickly to temperatures rising at higher temperatures than at lower temperatures [10]. Microbial communities can be structured by the interactions between geochemical conditions and microbial capabilities mechanistically [11]. Several studies have shown that the diversity of microbial communities is influenced by some soil properties, such as pH [12], organic carbon [13], nitrate [14], nitrite [15], ammonium [16], and phosphate [17]. Therefore, microbial diversity and structure changes may reflect environmental changes in the study area to some extent.

In the aspect of microbial community diversity, from the study of Chong et al. [18], the distribution patterns of soil bacteria in different regions of Antarctica were summarized. At a higher taxonomic level, such as phylum and class, the soil bacterial community composition can be highly stable while at lower taxonomic levels, it might be sensitive to both spatial and environmental gradients. Bacteroides were the most widespread phyla (89%) of the study sites followed by actinomycetes (86%) and acid bacteria (77%). Regional community differences may be caused by latitude, climate, and geological characteristics. In sediments of West Lake Bonney, McMurdo Dry Valleys, Bacteriodetes and Proteobacteria were found to be the major bacterial phyla while Thaumarchaeota and Crenarchaeota were the main archaea [19]. Firmicutes, Deltaproteobacteria, and Epsilonproteobacteria were identified as the dominant bacteria, and Thaumarchaeota was the major archaea in cryoconite ecosystems [20]. Bacteroidetes, Acidobacteria, Betaproteobacteria, Alphaproteobacteria, and Actinobacteria were the dominant phyla in soil samples from the Keller Peninsula, King George Island, Antarctica [21]. Given that the Kitezh Lake is the largest lake on the Fildes Peninsula, King George Island, Antarctica, and it is far from the glacier and less affected by glacier melt water, which makes the lake volume and intertidal area relatively stable, it was chosen to be the research area to study the diversity and composition of bacterial and archaeal communities in soils.

To obtain the relationship between microbial communities and various environmental factors, in traditional research methods, redundancy analysis (RDA) or canonical correspondence analysis (CCA) analysis is often used. However, it has been pointed out in some literature that either RDA or CCA has certain limitations [22]. As the second coordinate axis is often affected by the first coordinate axis, that is, the existence of the bow effect [23], it cannot correctly reflect the differences and relationships among variables. We tried to use other methods to explore the relationship between microbial communities and geochemical factors accurately. Weighted gene coexpression network analysis (WGCNA) is an R package which was originally used to search for key genes of some diseases and traits [24,25,26]. Due to the similarity between operational taxonomic units (OTUs) and genes, some researchers have applied it to the microbiome to find hub OTUs, such as those of the rhizosphere [27] and potato soils [28]. Hub OTUs refers to a series of OTUs that have high connectivity and a high degree of coexpression with others.

Therefore, we hypothesized that the bacterial and archaeal community diversity and composition in three types of soil were influenced by eight soil geochemical properties: pH, moisture content, and total carbon are important properties of soils; nitrate, nitrite, ammonium, and phosphate are often limiting nutrients in poor nutrition areas; and silicate is a ubiquitous nutrient and can also affect microbial communities when added to soil as fertilizer [29]. The competing hypothesis is that not all of these eight soil geochemical properties play a role. The content of this study mainly includes two aspects. On the one hand, the diversity and composition of bacterial and archaeal communities in three different types of soils in the Kitezh Lake area were analyzed. In addition, on the other hand, the correlation between the geochemical properties of the soil and bacterial and archaeal community diversity and composition was revealed by RDA/CCA and WGCNA. Hub microbiota correlated with geochemical properties were found. The advantages and disadvantages of the two methods were analyzed.

## 2. Materials and Methods

### 2.1. Study Sites and Sample Collection

The study site was the Kitezh Lake which is located in the middle of the Fildes Peninsula of King George Island in Antarctica. This area has a warming trend, is relatively humid [30], and belongs to the sub-Antarctica maritime climate. The vegetation in this area is very sparse. There are little Deschampsia antarctica and mosses in the sampling area but not in the sampling site. The microorganism quantity is small and mainly in the soil, so microbial mats will not form. Only a small number of Pygoscelis adeliae and seals are found locally, mainly along the coast, with little animal activity at sampling sites near the lake. Samples were collected in February 2018 for three sites, including BPK (below-peak soil), INT (intertidal soil), and SED (sediment) (Figure 1). Fifty grams of topsoil (0–5 cm) was collected into TWIRL’EM sterile bags (Labplas Inc., Sainte-Julie, QC, Canada) using a sterile shovel for each sample. Three soil samples were taken from each site for a total of nine samples. Once collected, samples were stored at −20 °C in the Great Wall Station (China) for 10 days before being transported to the home laboratory in a cooler. Then, they were stored at −80 °C until DNA extraction and analysis of geochemical properties.

### 2.2. Geochemical Properties of Soils

The moisture content (MC), pH, total organic carbon (TOC), and concentrations of five soluble nutrients, including nitrate (NO_3_^−^-N), nitrite (NO_2_^−^-N), ammonium (NH_4_^+^-N), silicate (SiO_4_^2−^-Si), and phosphate (PO_4_^3−^-P), of the samples were measured. The moisture content of soil samples was determined by the drying method, that is, soil samples were dried to a constant weight at 105 °C, and then the proportion of water loss was calculated. Ten grams of soil samples were freeze-dried and ground into powder in which gravel and roots were removed. After soil samples were treated with 10% HCl and dried, TOC was analyzed using an element analyzer (EA30000, Euro Vector SpA, Milan, Italy). Soil pH was measured by adding 10 mL of distilled water to 4 g of soil using a pH meter (PHS-3C, Shanghai REX Instrument Factory, Shanghai, China). The soil samples used for nutrient content measurement were also freeze-dried and ground, and then water was added at a ratio of 1:10 (g·mL^−1^). After shaking once every 4 h for 48 h, a nutrient autoanalyzer (SEAL Analytical Gmbh, QuAAtro, Tastrup, Germany) was used to assess the dissolved nutrients, including ammonium (NH_4_^+^-N), nitrate (NO_3_^−^-N), nitrite (NO_2_^−^-N), silicate (SiO_4_^2−^-Si), and phosphate (PO_4_^3−^-P) with a standard deviation of 5% and detection limits of 0.040, 0.015, 0.030, 0.030, and 0.024 μmol/L, respectively.

### 2.3. DNA Extraction, PCR Amplification, and Products Processing

DNA was extracted from a 0.25 g soil sample using a Power Soil DNA Isolation Kit (MO BIO Laboratories, San Diego, CA, USA) according to the manufacturer’s instructions. The integrity and purity of DNA obtained were detected on an agarose gel. DNA concentrations were measured using the Qubit dsDNA Assay Kit in Qubit 2.0 Fluorometer (Life Technologies, Carlsbad, CA, USA). Polymerase chain reaction (PCR) amplification of bacteria was performed using primers 341F (5′-CCTAYGGGRBGCASCAG-3′) and 806R (5′-GGACTACNNGGGTATCTAAT-3′) [31] for the V3 and V4 regions of 16S rRNA gene. All PCR reactions were carried out in 30 μL reactions including 15 μL of Phusion High-Fidelity PCR Master Mix (New England Biolabs, Ipswich, MA, United States), 0.2 μmol·L^−1^ of forward and reverse primers, and 10 ng template DNA. The PCR amplification cycle was initial denaturation at 98 °C for 1 min followed by 30 cycles of denaturation at 98 °C for 10 s, annealing at 50 °C for 30 s, elongation at 72 °C for 30 s, and a final extension of 72 °C for 5 min. The differences between archaea and bacteria PCR amplification mainly lie in the use of primers and the amplification cycle. The V4 and V5 regions of the archaea 16S rRNA gene were amplified using the primers Arch519F (5′-CAGCCGCCGCGGTAA-3′) and Arch915R (5′-GTGCTCCCCCGCCAATTCCT-3′) [32]. In addition, the PCR amplification cycle was as follows: initial denaturation at 94 °C for 4 min followed by 30 cycles of denaturation at 94 °C for 15 s, annealing at 56 °C for 30 s, and elongation at 68 °C for 80 s with a final extension of 72 °C for 10 min.

The products were mixed with an equal volume of 1× loading buffer containing SYB green and then loaded onto 2% agarose gel for detection. Samples with an amplification signal between 400 and 450 bp were chosen for purification with Gene JET Gel Extraction Kit (Thermo Scientific, Waltham, MA, USA).

### 2.4. Sequence Analysis

Each sample’s data were separated from the off-machine data according to barcode sequence and PCR amplification primer sequence. After cutting off the barcode and primer sequence, FLASH (V1.2.7) was used to splice reads of each sample, and the resulting spliced sequence was the raw reads. The raw reads were deposited into the NCBI Sequence Read Archive database (accession number: SRP381813).

The 16S rRNA gene amplicons were sequenced on the Illumina MiSeq platform [33], and 250 bp paired-end reads were generated. Clean reads were obtained using QIIME (Version 1.8.0) to filter low-quality bases and then compared with species annotation database to detect and remove the chimeric sequences. The Uparse software [34] (Version 7.0) was used for clustering the clean reads of all samples into OTUs at a 97% identity. The representative sequence was the most frequent sequence for each OTU. Species were annotated and analyzed on the representative sequences of OTUs using QIIME and the Silva rRNA database [35] (Version 138) to obtain taxonomic information and calculate the abundance at each taxonomic level in all samples.

### 2.5. Statistical Analysis

The alpha diversity of three study sites was compared in terms of ACE, Chao1, Shannon’s index (H’), Simpson diversity index, and Good’s coverage using QIIME 1.8.0. The species accumulation boxplot was plotted using R v4.0.5 to check if the samples were sufficient. To find key geochemical factors affecting community composition of bacteria and archaea in soil, detrended correspondence analysis (DCA), CCA, and RDA were performed in R. Through the results of DCA, when the axis length of DCA1 (the first axis length of DCA) is smaller than 3, RDA is better than CCA; when it is bigger than 4, CCA is preferred; and when it is between 3 and 4, both RDA and CCA can be selected. All 8 environmental variables were included in CCA and RDA. Correlation and significance analyses, such as one-way analysis of variance (ANOVA) and bivariate correlation analysis (Spearman’s correlation analysis [36]), were conducted using IBM SPSS Statistics 26.

The R package WGCNA [37], which is a weighted coexpression network analysis method, was used to identify relationships between geochemical factors and the microbiome. The process of WGCNA is to first calculate the correlation coefficient (Pearson correlation) between any two OTUs, and it is power-weighted to make the connection between OTUs in the network obey the scale-free network distribution, classifying the OTUs based on the weighted correlation coefficient of OTU, and the OTUs with similar distribution were classified into one module. Then, the relationship between the geochemical factors and modules was analyzed to find modules that have significant correlations (Pearson correlation) with geochemical factors. Then, these modules were exported, analyzed, and visualized in Cytoscape 3.8.1. The top 200 OTUs with the highest weight in each module were chosen to draw the network diagram. Finally, through the calculation and comparison of weight values of OTUs in the modules, the hub OTUs were obtained, which refer to the OTUs with the highest connectivity and high degree of coexpression with other OTUs in the modules. The results of RDA and WGCNA were compared to analyze the advantages and disadvantages of the two analysis methods.

## 3. Results

### 3.1. Geochemical Properties of Soil Samples

Eight geochemical properties were measured: MC, pH, TOC, NO_3_^−^-N, NO_2_^−^-N, NH_4_^+^-N, SiO_4_^2−^-Si, and PO_4_^3−^-P. As can be seen from Table 1 and Appendix A, differences were observed at different sites. The concentration of SiO_4_^2−^-Si increased from the bank slope to the sediments while NO_2_^−^-N decreased. The other six geochemical properties were all detected at the lowest values at the INT sites. Among them, MC, NH_4_^+^-N, NO_3_^−^-N, and PO_4_^3−^-P increased from the bank slope to the sediments, and the rest were reversed.

### 3.2. Diversity and Structure Analysis of Bacterial and Archaeal Communities

In terms of bacteria, a total of 645,661 bacterial sequences and 1278 OTUs were identified from the nine samples of the three sites. The alpha diversity index of the bacteria is shown in Appendix A and Appendix A. Chao1 and ACE are parameters used to describe the richness of microbiota. It can be seen that the richness of INT was the highest and that of SED was the lowest. The Shannon index indicated that INT had the highest homogeneity among the three sites, and the Simpson index indicated that INT had the largest species diversity and SED had the least. The Good’s coverage ranged from 0.9977 to 0.9997, indicating that the sequences sufficiently covered most of the bacterial populations in all samples. The species accumulation boxplot (Appendix A) shows that sample capacity was sufficient.

As for archaea, a total of 139,988 archaeal sequences and 108 OTUs were obtained. The alpha index of the archaea is listed in Appendix A and Appendix A. Different from the bacteria, the ACE and Chao1 values of the archaea reached the highest at the SED site, though Sed1-2 values were relatively low. The Shannon and Simpson indices of the archaea were also different from those of the bacteria, and there was little difference among the three sites, indicating that the archaeal communities in the three sampling sites had similar homogeneity and diversity. The Good’s coverage ranged from 0.9957 to 0.9998, indicating that the sequencing depth is sufficient. The species accumulation boxplot can be seen in Appendix A.

As is shown in Figure 2, the taxonomy data covered 23 identified phyla, of which, Proteobacteria (84.06% in BPK, 43.25% in INT, 96.53% in SED; full results can be seen in Appendix A), Actinobacteria (8.86%, 26.58%, 1.73%), Bacteroidetes (1.31%, 10.54%, 0.51%), Acidobacteria (0.04%, 5.79%, 0.13%), and Firmicutes (4.62%, 0.36%, 0.72%) were the top five phyla with the highest bacterial abundance (Appendix A). The relative abundance of Proteobacteria in samples of BPK and SED exceeded 80%, and those in samples of INT also reached 40%, occupying the main dominance. Among phyla in which the relative abundance results conformed to the homogeneity of variance, Proteobacteria got the highest relative abundance in SED and Actinobacteria had a higher abundance in INT while the relative abundance of Firmicutes was the highest in BPK. In addition, at the genus level, 41 classifiable genera were identified while *Ralstonia* (45.64%, 2.55%, 37.73%), *Sphingomonas* (18.51%, 6.39%, 50.11%), *Rhodococcus* (8.47%, 0.50%, 1.59%), *Phyllobacterium* (4.71%, 0.28%, 3.42%), *Oryzihumus* (0.09%, 10.67%, 0.02%), *Bradyrhizobium* (2.83%, 1.06%, 1.63%), *Flavobacterium* (0.02%, 7.64%, 0%), *Delftia* (2.38%, 0.14%, 1.65%), *Sphingopyxis* (0.05%, 6.93%, 0%), and *Bosea* (4.21%, 0.14%, 0.05%) were the top 10 genera (Appendix A). Of the two genera with the highest abundance, *Ralstonia* had a higher relative abundance in BPK and INT while *Sphingomonas* was the most abundant bacterial genera in SED. The relative abundance of *Bradyrhizobium* was higher in BPK and SED. In addition, in general, the three sites of INT had a greater diversity of species than others no matter the phylum or genus level.

In Figure 3, it can be seen that all OTUs of the archaea in the nine samples were mainly divided into 10 genera of seven phyla except for the unclassified ones. The phylum with the highest relative abundance was Crenarchaeota, which accounted for more than 80% at each site (83.46%, 84.56%, 84.13%). At the level of genus, the relative abundance of *Candidatus Nitrososphaera* (29.81%, 29.61%, 26.05%) and *Candidatus Nitrocosmicus* (0.21%, 1.01%, 2.48%) was greater than that of others (Appendix A).

### 3.3. Correlation between Geochemical Properties and Bacterial and Archaeal Community

CCA was performed to explore the relationship between geochemical factors and bacterial community (Figure 4). The first two axes explained 80.19% of the total variance in the bacterial community structures. It can be seen that INT was clearly distinguished from the other two groups of samples, and all eight soil geochemical properties were negatively correlated with the sample distribution. In addition, according to a Monte Carlo permutation test (Table 2), the concentrations of PO_4_^3−^-P (*r* = 0.8631, *p* < 0.01) and pH (*r* = 0.7529, *p* < 0.05) were the two most important factors associated with bacterial community composition.

As the axis length of DCA1 was smaller than three, RDA was used to explore the correlation between geochemical factors and the archaeal community (Figure 5). The first two axes explained 99.98% of the total variance in the archaeal community structures. As is shown in Table 3, pH (*r* = 0.6987, *p* < 0.05) and NO_2_^−^-N (*r* = 0.6699, *p* < 0.05) played a significantly important role in the archaeal community composition.

To identify microbial populations with differences in distribution between sample sites, a one-way ANOVA analysis with a homogeneity test of variances was conducted on the top 10 phyla and top 20 genera in bacteria abundance, and the Waller Duncan test was used post hoc to mark the phyla and genera identified. As a result, four phyla and six genera were selected (Appendix A). Then, Appendix A was obtained by Spearman correlation analysis of bacterial phyla or genera and geochemical properties. As is shown in Appendix A, at the level of phylum, Firmicutes (*r* = 0.9, *p* < 0.01) was positively correlated with pH while Acidobacteria (*r* = −0.917, *p* < 0.01), Gemmatimonadetes (*r* = −0.733, *p* < 0.05), and Saccharibacteria (*r* = −0.762, *p* < 0.05) were negatively correlated with it. Gemmatimonadetes (*r* = −0.667, *p* < 0.05) and Saccharibacteria (*r* = −0.678, *p* < 0.05) were negatively correlated with NH_4_^+^-N. Four phyla were significantly correlated with PO_4_^3−^-P, which were Proteobacteria (*r* = 0.817, *p* < 0.01), Actinobacteria (*r* = −0.817, *p* < 0.01), Bacteroidetes (*r* = −0.833, *p* < 0.01), and Nitrospirae (*r* = −0.797, *p* < 0.01). In addition, Firmicutes was found to have a significant correlation with NO_2_^−^-N (*r* = 0.717, *p* < 0.05). Moreover, at the genus level, PO_4_^3−^-P was the most influential factor, as 10 genera were significantly correlated with it (*p* < 0.05), including *Sphingomonas*, *Oryzihumus, Sphingopyxis*, etc. NH_4_^+^-N and pH were the next two influential factors, which were correlated with six and four genera, respectively. A total of three and two genera correlated with NO_2_^−^-N and SiO_4_^2−^-Si. In terms of archaea, the results of the one-way ANOVA analysis showed that there was no significant difference in abundance distribution between sampling sites (Appendix A). The Spearman correlation analysis was performed for dominant phyla and genera and geochemical properties (Appendix A); except for the unclassified ones, one phylum and one genus were identified. Crenarchaeota had a significant negative correlation with pH (*r* = −0.8, *p* < 0.05). In addition, at the genus level, *Candidatus Nitrocosmicus* was negatively correlated with pH (*r* = −0.8, *p* < 0.01) and NO_2_^−^-N (*r* = −0.833, *p* < 0.01).

As for the network analysis, it can be seen in Figure 6a that among the nine bacterial modules, four modules were significantly correlated with geochemical properties: module magenta had a negative correlation with moisture content (*r* = −0.76, *p* < 0.05), module green had a negative correlation with pH (*r* = −0.66, *p* < 0.05), module blue had negative correlations with NH_4_^+^-N (*r* = −0.66, *p* < 0.05), and module pink had positive correlations with TOC (r = 0.83, *p* < 0.01) and NO_2_^−^-N (*r* = 0.91, *p* < 0.01). At the same time, from Figure 6b, among the six archaeal modules, two modules were significantly correlated with geochemical properties: module turquoise with NO_3_^−^-N (*p* < 0.05) and module blue with MC (*p* < 0.05). Figure 7 contains the network diagrams of the four bacteria modules and two archaea modules mentioned above. The hub OTUs of these modules are listed in Table 4.

Through CCA and RDA, pH and PO_4_^3−^-P were the strongest environmental factors influencing the diversity of the bacterial community while pH and NO_2_^−^-N were the two strongest influences for the archaeal community. However, in the WGCNA results, MC, NH_4_^+^-N, NO_3_^−^-N, and TOC were also found to be geochemical properties that affect the diversity and structure of bacterial or archaeal communities.

## 4. Discussion

As temperatures often rise above the melting point of ice in summer [38], the water level and area of the Antarctic lake region rise, dividing the lake’s soils into slope soils (above water all year round), intertidal soils (underwater in summer, above water in winter), and sedimentary soils (submerged by water/ice all year round). Based on this phenomenon, we hypothesized that the bacterial and archaeal community diversity and composition in these three types of soil were influenced by soil geochemical properties. Our study aimed to explore differences in the diversity and composition of bacterial and archaeal communities at different sites from below-peak soil (slope soil) to intertidal soil and sediment and to find the correlation between these differences and geochemical properties. The study by E. Yergeau and G. A. Kowalchuk [39] verified that following global warming, increases in average temperature and the frequency of freeze-thaw cycles may influence Antarctic soil microbial communities, and vegetation may influence some soil environmental parameters, such as through N-cycles, and then influence the soil microbes. Similarly, the sampling sites we chose could be seen as different stages of freeze-thaw cycles, and nitrate, nitrite, and ammonium were all important geochemical factors in our study. It is also reported that the oxic layers of the lake corresponded with nitrate while the deep anoxic layers corresponded with NH_4_^+^ [40]. In our study, we can draw a similar conclusion: bacterial module blue is correlated with NH_4_^+^-N, and the hub OTUs in this module are highly abundant in INT and less abundant in SED, but SED has a higher NH_4_^+^-N concentration in these two sampling sites.

In terms of diversity analysis of the bacteria and archaea, phyla and genera with high abundance were found. The dominant phyla of the bacteria are Proteobacteria, Actinobacteria, and Bacteroidetes, which is consistent with previous studies [41,42,43]. Through the Spearman analysis, among the phyla with a significant correlation with pH, except for Firmicutes, the other three phyla had positive correlations with decreasing pH and higher abundance in INT. Firmicutes had a higher abundance in BPK and had a positive correlation with alkaline pH. In previous studies, Firmicutes is found to exist in the alkaline or slightly acidic soil in the Ross Sea region [44] and Marble Point and Wright Valley, Victoria Land [45] in Antarctica. At the genus level, among those 18 genera which have a different distribution across the sampling sites, 12 genera had higher abundance in INT. This may reflect the greater bacterial richness of the INT from another aspect. No matter the phylum or genus classification, PO_4_^3−^-P is an influential geochemical property, which is similar to the results of other research [46,47,48]. According to the Spearman analysis (Appendix A), *Sphingomonas*, which has the second highest abundance at the genus level, was found to have a correlation with phosphate (*r* = 0.949, *p* < 0.01). Some *Sphingomonas* strains were found to have the function of solubilizing phosphate [49,50], which may support the result of CCA that phosphate was an influential geochemical property for the bacterial community. As for the archaea, the identified dominant phylum of archaea is Crenarchaeota (84.05%). It was also reported as the major archaea of the Ross Sea region [51], Helliwell Hills [52], and Dry Valleys [53] in Antarctica. Crenarchaeota is often found to be a major archaeal phylum that is anaerobic, thermophilic, and acidophilic [54] and includes widely existing ammonia-oxidizing archaea [55,56]. *Candidatus Nitrososphaera* [57] and *Candidatus Nitrocosmicus* [58] were found to be the dominant genera, also indicating that ammonia-oxidizing archaea were widespread in this region to participate in the nitrogen cycle process. This could also explain why nitrite was identified as a key factor in the structure and composition of the archaea community.

Through WGCNA of bacteria and archaea, in addition to pH, PO_4_^3−^-P, and NO_2_^−^-N, MC, NH_4_^+^-N, NO_3_^−^-N, and TOC were also found to be influential factors. In addition, they were also supported by research in Antarctica [13,59,60]. On the one hand, combined with the OTU table and geochemical properties of the sampling sites, we found relationships between geochemical properties and OTU abundance. On the other hand, the values of the geochemical properties of the nine sampling sites can verify our results from another aspect. For example, Bpk1-2 was correlated with TOC and NO_2_^−^-N positively, and the two values of Bpk1-2 were the highest among the nine sites. Based on the above results, the alternative hypothesis is accepted.

As for the analysis process, the traditional method is to get the community richness and diversity by alpha diversity analysis, compare the differences between groups by beta diversity analysis, acquire high relative abundance taxonomies by community structure analysis, obtain the geochemical properties that play a key role in the distribution of microbial diversity through RDA or CCA, and finally, find the significant microbiota and geochemical factors by calculating the correlation between them. In this study, modules that have strong correlations with geochemical properties were obtained through WGCNA, and the hub OTUs in these modules were searched to obtain their correlations with geochemical factors. Both methods can obtain highly correlated microbes and geochemical properties, and then combined with the distribution of abundance, we can analyze the impact of geochemical factors on sampling sites. However, due to the arch effect of RDA [23], we believe that there may be some data inaccuracies. In addition, the explanation of the first two axes may be low sometimes; it is also a disadvantage of RDA for being not convincing. However, WGCNA does not have these problems. This method obtains the hub OTUs through the calculation of adjacency and connectivity and can intuitively find the hub microbiotas related to geochemical factors. In a word, the method of WGCNA is less time consuming and can find the key microbes more efficiently, which is helpful to provide a reference for the further development of microbial resources. However, our results and discussion are based on a relatively small sampling number, which may be a limitation of this study. Limitations also lie in the phenotypic variability within a phylum and collinearity between environmental variables. In future research, we can increase the sample size to obtain more accurate statistical data, analyze the correlations between environmental variables and microbial community structure at the lower level, and discuss the effect of collinearity between environmental variables.

## 5. Conclusions

In this study, we found differences in bacterial and archaeal community structures in the bank slope soil, intertidal soil, and sediment of Kitezh Lake in Antarctica. CCA and RDA showed that pH and phosphate had significant effects on the bacterial community while the archaeal community was mainly affected by pH and nitrite. By WGCNA, moisture content, ammonium, nitrate, and total carbon content were also found to be influential geochemical properties. The hub bacteria and archaea that were significantly correlated with geochemical properties were obtained, which is more intuitive and specific. This work provides a new idea for future microbial diversity analysis and contributes to the change of microbial communities in polar lake regions under the aggravation of global warming.

## Figures and Tables

**Figure 1 biology-11-01855-f001:**
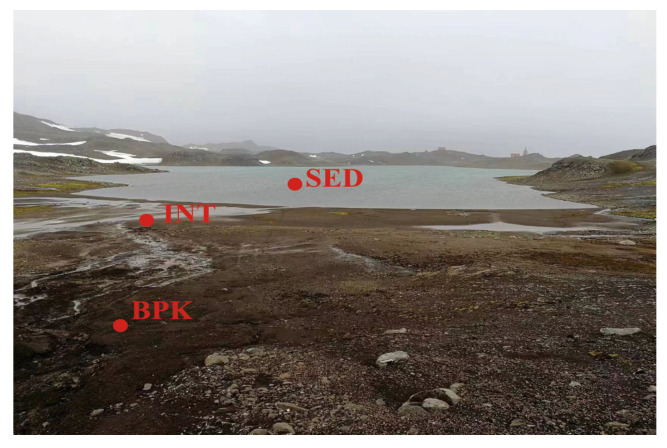
Sampling sites in the Kitezh Lake area.

**Figure 2 biology-11-01855-f002:**
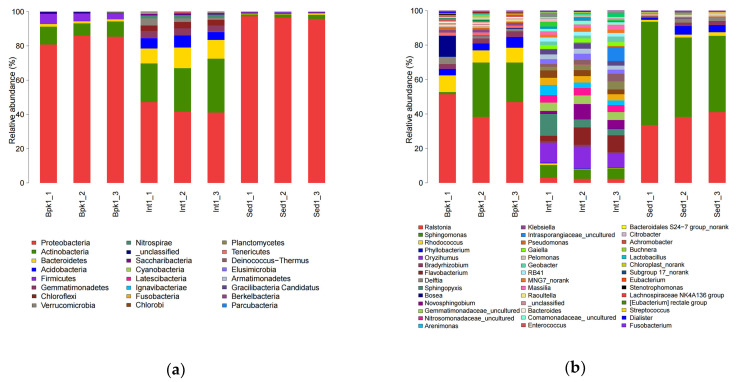
Bar charts of relative species abundance of bacteria at the phylum level (**a**) and genus level (**b**).

**Figure 3 biology-11-01855-f003:**
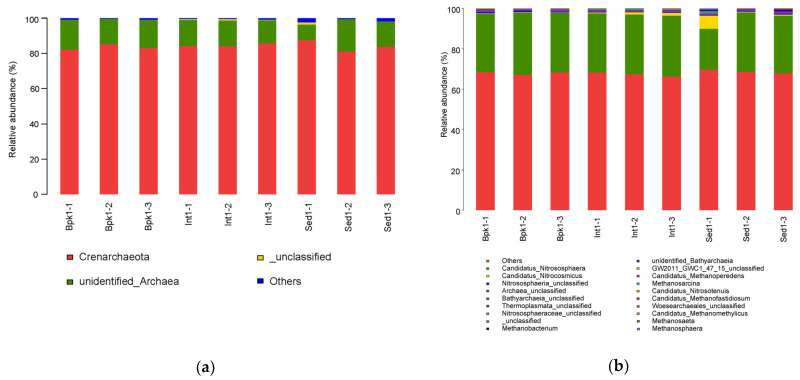
Bar charts of relative species abundance of archaea at the phylum level (**a**) and genus level (**b**).

**Figure 4 biology-11-01855-f004:**
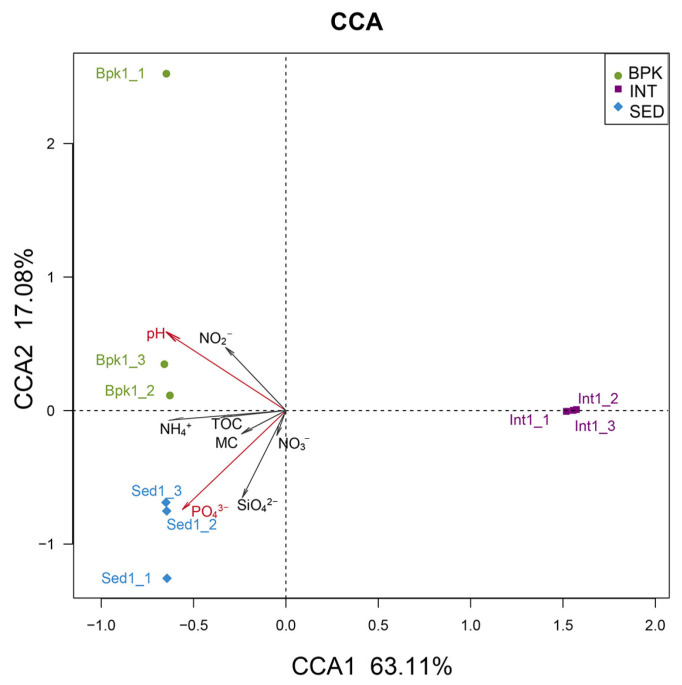
Canonical correspondence analysis of bacterial community. Dots with different colors represent different sampling sites, arrows represent geochemical properties, and the longer the length, the greater the correlation between the geochemical property and the sample distribution.

**Figure 5 biology-11-01855-f005:**
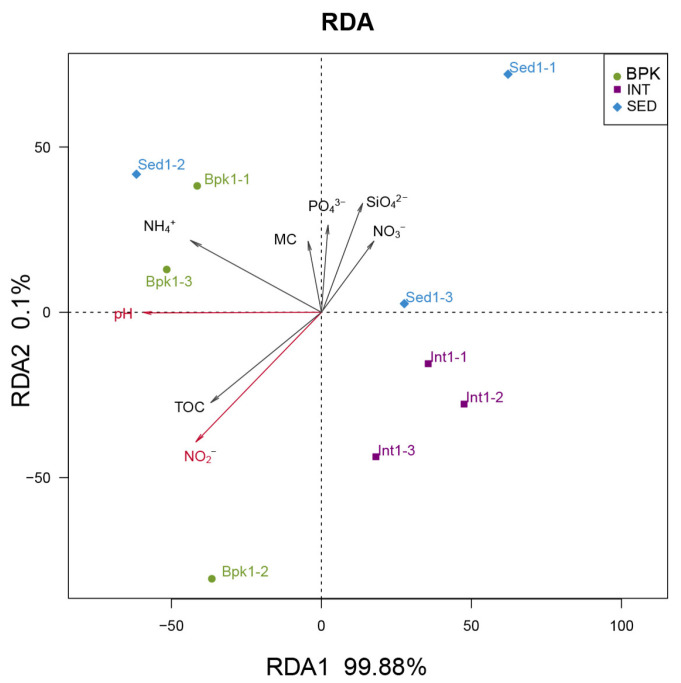
Redundancy analysis of archaeal community. Dots with different colors represent different sampling sites, arrows represent geochemical properties, and the longer the length, the greater the correlation between the geochemical property and the sample distribution.

**Figure 6 biology-11-01855-f006:**
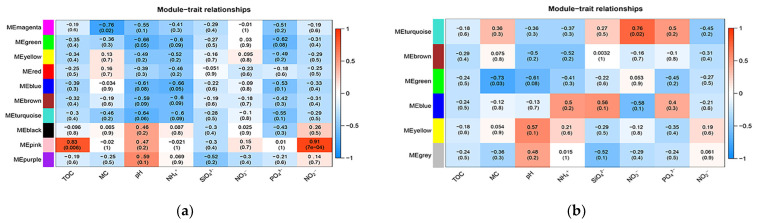
Module–geochemical property relationships of bacteria (**a**) and archaea (**b**). The abscissa is the geochemical properties, and the ordinate is the modules. The red squares show a positive correlation, and the green ones show a negative correlation. The darker the color, the stronger the correlation.

**Figure 7 biology-11-01855-f007:**
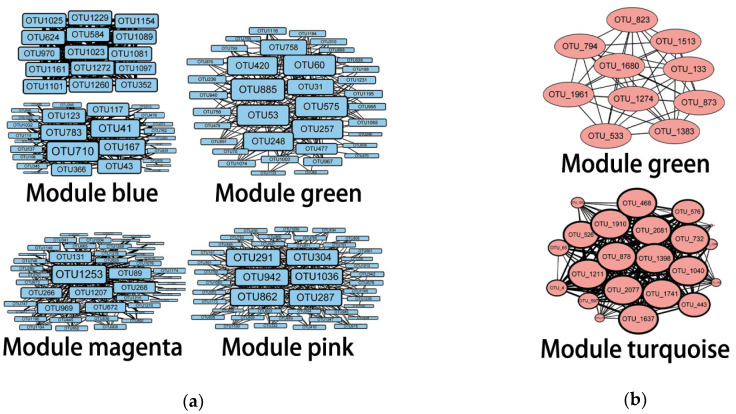
The network diagrams of bacteria (**a**) and archaea (**b**). The nodes represent OTUs, and the lines represent the connections between them. The bigger the size, the stronger the correlation of this OTU with others, and the larger the weight value of the OTU is to identify it as the hub OTU. A total of 200 OTUs were used to draw the network diagrams, which resulted in two networks in module blue of (**a**). However, they will eventually be connected through the OTUs with a relatively low weight value.

**Table 1 biology-11-01855-t001:** Geochemical properties of sampling sites.

Site		MC (%)	TOC (%)	pH	NH_4_^+^-N (μg/g)	SiO_4_^2−^-Si (μg/g)	NO_3_^−^-N (μg/g)	PO_4_^3−^-P (μg/g)	NO_2_^−^-N (μg/g)
BPK	Bpk1-1	22.55	0.024	7.69	0.1852	15.6632	0.5687	0.4938	0.0323
	Bpk1-2	22.01	0.138	7.62	0.1595	16.3551	0.6059	0.6860	0.0519
	Bpk1-3	20.13	0.010	7.77	0.1774	13.3118	0.4997	0.5875	0.0269
	Average	21.56 ± 1.43 ^a^	0.057 ± 0.047 ^a^	7.69 ± 0.08 ^a^	0.1740 ± 0.0145 ^a^	15.1100 ± 1.7982 ^b^	0.5581 ± 0.0584 ^a^	0.5891 ± 0.0969 ^b^	0.037 1± 0.0148 ^a^
INT	Int1-1	24.80	0.004	6.91	0.0960	18.5088	0.5979	0.5165	0.0178
	Int1-2	14.95	0.022	6.74	0.1063	16.0028	0.5613	0.4836	0.0202
	Int1-3	23.97	0.015	6.96	0.0966	19.3438	0.5076	0.6380	0.0185
	Average	21.24 ± 6.29 ^a^	0.014 ± 0.010 ^a^	6.87 ± 0.13 ^b^	0.0996 ± 0.0067 ^a^	17.9518 ± 1.9490 ^b^	0.5556 ± 0.048 ^a^	0.5460 ± 0.0920 ^b^	0.0188 ± 0.0014 ^b^
SED	Sed1-1	25.18	0.016	6.92	0.1037	22.8561	0.7280	0.8692	0.0113
	Sed1-2	23.90	0.085	7.20	0.3028	26.7257	0.5692	0.8407	0.0203
	Sed1-3	20.91	0.012	7.09	0.2597	26.1310	0.4309	0.8116	0.0196
	Average	23.33 ± 2.42 ^a^	0.038 ± 0.047 ^a^	7.07 ± 0.15 ^b^	0.2221 ± 0.1184 ^a^	25.2376 ± 2.3815 ^a^	0.5761 ± 0.1519 ^a^	0.8405 ± 0.0289 ^a^	0.0171 ± 0.0058 ^b^

Significant differences between the study sites in one-way ANOVA at *p* < 0.05 followed by the Duncan test. The letters ^a^ and ^b^ indicate significant differences between study sites.

**Table 2 biology-11-01855-t002:** A Monte Carlo permutation test for geochemical properties and bacterial community.

	CCA1	CCA2	*r* ^2^	*p*	
TOC	−0.98939	−0.14529	0.1234	0.647	-
MC	−0.81181	−0.58392	0.0833	0.782	-
pH	−0.74304	0.66925	0.7529	0.026	*
NH_4_^+^	−0.99353	−0.11355	0.4000	0.207	-
SiO_4_^2−^	−0.34077	−0.94015	0.4789	0.142	-
NO_3_^−^	−0.25634	−0.96659	0.0369	0.869	-
PO_4_^3−^	−0.60339	−0.79744	0.8631	0.004	**
NO_2_^−^	−0.56632	0.82419	0.3120	0.284	-

* Correlation is significant at the 0.05 level. ** Correlation is significant at the 0.01 level; *p*-values based on 999 permutations.

**Table 3 biology-11-01855-t003:** A Monte Carlo permutation test for geochemical properties and archaeal community.

	RDA1	RDA2	*r* ^2^	*p*	
TOC	−0.80330	−0.59557	0.4289	0.205	-
MC	−0.20201	0.97938	0.0977	0.735	-
pH	−0.99999	−0.00319	0.6987	0.035	*
NH_4_^+^	−0.89564	0.44478	0.4669	0.160	-
SiO_4_^2−^	0.42833	0.90362	0.1851	0.547	-
NO_3_^−^	0.63076	0.77598	0.1569	0.643	-
PO_4_^3−^	0.08396	0.99647	0.1425	0.619	-
NO_2_^−^	−0.72962	−0.68386	0.6699	0.029	*

* Correlation is significant at the 0.05 level; *p*-values based on 999 permutations.

**Table 4 biology-11-01855-t004:** Hub OTUs of bacteria and archaea.

Module	OTU Number	Assignment	Taxon
Bacterial module blue	OTU710	Gaiellales	order
	OTU41	Actinobacteria MB-A2-108	class
	OTU783	Unclassified	-
	OTU167	*Sphingomonas*	genus
	OTU123	Betaproteobacteria SC-I-84	order
	OTU43	*Blastocatellaceae RB41*	genus
	OTU1101	*Eisenbergiella*	genus
	OTU1154	*Ruminococcaceae UCG-014*	genus
	OTU1260	*Brevundimonas*	genus
	OTU584	Unclassified	-
	OTU1272	*Roseburia*	genus
	OTU1161	*Deinococcus*	genus
	OTU1229	Berkelbacteria	phylum
	OTU1081	*Ruminiclostridium 5*	genus
	OTU1025	*Fusobacterium*	genus
	OTU1023	*Ruminococcaceae UCG-005*	genus
	OTU1089	*Ruminococcaceae UCG-014*	genus
	OTU624	*Lachnospiraceae NK4A136 group*	genus
Bacterial module green	OTU885	*Nitrospira*	genus
	OTU60	Acidimicrobiaceae	family
	OTU53	*Ferruginibacter*	genus
	OTU575	*Segetibacter*	genus
	OTU257	Verrucomicrobia OPB35 soil group	class
Bacterial module magenta	OTU1253	*Desulfovibrio*	genus
	OTU1207	Betaproteobacteria SC-I-84	order
	OTU89	Chthoniobacterales DA101 soil group	family
	OTU131	Chloroflexi KD4-96	class
	OTU969	Verrucomicrobiaceae	family
	OTU268	*Hyphomicrobium*	genus
	OTU266	Holophagae Subgroup 7	order
	OTU672	Solibacteraceae	family
Bacterial module pink	OTU291	Xanthomonadales	order
	OTU862	*Anaeromyxobacter*	genus
	OTU942	Unclassified	-
	OTU287	*Arsenophonus*	genus
	OTU304	*Lachnospiraceae NK4A136 group*	genus
	OTU1036	*Lachnospiraceae NK4A136 group*	genus
Archaeal module green	OTU_823	Nitrososphaeria Group_1.1c	order
	OTU_794	Unknown	-
	OTU_1513	Unknown	-
	OTU_1680	Woesearchaeales	order
	OTU_133	Nitrososphaeria	class
	OTU_1961	Unknown	-
	OTU_1274	Unknown	-
	OTU_873	Unknown	-
	OTU_533	Unknown	-
	OTU_1383	Woesearchaeales	order
Archaeal module turquoise	OTU_468	Unknown	-
	OTU_1910	Unknown	-
	OTU_2081	Methanomassiliicoccaceae	family
	OTU_732	Archaea	kingdom
	OTU_878	Micrarchaeales CG1-02-32-21	family
	OTU_1398	Woesearchaeales GW2011_GWC1_47_15	family
	OTU_1040	Woesearchaeales	order
	OTU_1211	Woesearchaeales	order
	OTU_2077	Woesearchaeales	order
	OTU_1741	Woesearchaeales	order
	OTU_1637	*Methanosphaera*	genus

## Data Availability

The data presented in this study are available upon request from the corresponding author.

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
