# Peer review of "Soil Geochemical Properties Influencing the Diversity of Bacteria and Archaea in Soils of the Kitezh Lake Area, Antarctica"

_biology, 2022, doi:10.3390/biology11121855_

Round 1
Reviewer 1 Report (Previous Reviewer 3)
I have reviewed the manuscript after the changes and agree with all answers provided by the authors. I recommend the publication as the manuscript presented in this final version.
Author Response
Thanks for your positive comment and wish you all the best.
Reviewer 2 Report (Previous Reviewer 2)
Number of samples are low and should be indicated as a limitation in method and discussion/conclusion sections.
Figures need to be regenerated. Currently, they are unclear/blurry.
Author Response
Thanks for your constrctive comment and our rivisions are as follows:
The following text has been added in the discussion section:”Our results and discussion are based on relatively small sampling number, which may be a limitation of this study. In future research, we can increase the sample size to obtain more accurate statistical data.”
Figures in the manuscript have been reinserted in the format of .eps in the hope of improving clarity.
Once again, thank you very much for your comments and suggestions. We hope that the revision is acceptable, and we look forward to hearing from you soon.
Reviewer 3 Report (New Reviewer)
This work basically searches for correlation between microbial diversity and environmental parameters (pH and nutrients) in Antarctic soils contrasted according to their water cover. However the authors claim to ‘investigate the effects of climate warming on microbial communities”, which has not been done in this study. The link between climate warming and environmental parameters (pH and nutrients) is not studied here, and the ecosystem effect (including the water cover) is highly overlooked and not statistically supported. I have several concerns regarding the impact and significance of this work, as well as methodological inconsistencies, that make it unsuitable for publication.
The introduction and literature analysis is very weak. Previous reports about microbial communities in Antarctic soils and sediments comparable to the ones studied here (and their link with environmental parameters) do exist: they are lacking here and must be included. The same occurs for the water table level effect. The paragraph about culture is out of the scope of this study (L73-82). An important problematic of this manuscript is the water cover extent (and its impact on microbial communities), but this rationale is not presented in the introduction. With the long paragraph on statistical methods (L62-73), it is not clear if this study will be a methodological evaluation and comparison of methods: this aspect is missing. The water cover may not be the only difference between sites. In particular, the vegetation cover, presence of microbial mats, and/or influence of local fauna in the sampling sites should be indicated and discussed.
Methodological concerns:
L115 : what is the limit of detection for nutrients quantification? How are the different analytical methods calibrated?
L121-128: It is surprising to use such a low annealing temperature for PCR. Do the authors have reference for that?. The risk is to have many unspecific hybridization at so low temperature: has it been checked?
L164-169: the definition of modules is not convincing, or at least not clearly explained: how is it related with networks? What is the “adjacency and similarity between OTUs”? It is not clear how/why the module definition can be based on “relationships with geochemical factors”, if afterwards the objective is to evaluate the correlation between module and geochemistry? It seems an endless circle.
Only correlations with 1 factor are shown: How are managed the multiple correlations with several parameters? And the possible collinearity between environmental variables? How are treated the inter-correlated variables? What are the quantitative criteria to define modules? For examples, modules ‘magenta’, ‘brown’, ‘turquoise’ and ‘green’ have similar patterns of correlation with environmental parameters, however they are divided in 4 modules: what was the criterion to do so? Usually modules are based on interactions between OTUs, without considering environmental parameters. The authors should clarify their approach.
An important conceptual/methodological problem is the use of “core” terminology, which has a conventional definition (i.e. the group of OTUs present in all the considered samples). In the present study, the core does not seem to follow this definition, can it be clarified? (indeed, L85: “we used WGCNA, which could find the core microbiota correlated with geo- 85 chemical properties”). If on the contrary it follows the conventional definition, important parameters have to be provided (thresholds of abundance and prevalence to include an OTU in the core). If it does not, the term has to be changed to a more appropriated one.
Results:
L2015-214: rather than the general taxonomic description provided here, the differences between the three sites (i.e. influence of water cover, and other parameters) should be better highlighted in terms of taxonomic composition. The abundance of the most important taxa (either most abundant, or identified as most critically differentiating the samples) should be given (average and standard deviation), and statistical tests should be done to test their significance between samples.
L225-232, Figure 4, Table2: it is not correct to show on the CCA plot the non-significant environmental variables (i.e. all variables except pH and phosphate). Same comment for Archaea plot.
CCA and RDA are constrained analysis (which explains the high (or very high) percentages of variance explained): it should be better stressed out here which variables have been included in the models, and thus which variables explain this amount of variance.
L240: what is DCA1? Where does the criterion ‘<3” comes from, what does it mean? The authors should explain the rationale beyond using RDA this time, and what it will provide that CCA did not. It is not clear why the authors chose one type of analysis for Bacteria and a different one for Archaea.
Correlation analysis: The objective of the authors is very confusing. The ANOVA analysis is intended to identify specific microbial taxa statistically associated with the geochemical factors (as done in Table 4). However, the authors state that their ANOVA post hoc test was used to identify the taxa with differences in distribution between sample sites, which is a completely different objective. It must be clarified. Regarding the Spearman correlation analysis, it is unclear while only 4 parameters are shown in Table 4, whereas 8 parameters are available and have been presented in the previous analysis. L256-276: all this section is a repetition of Table 4, without added value.
Network analysis: The understanding of the Results is strongly impaired by the lack of clear methodological description, as previously commented. 10 bacterial modules and 6 archaeal modules are presented in Figure 6; however only 4 bacterial modules and 2 archaeal modules are described in the text (L281-288) and further studied at the OTU level in the following figure 7: why, and upon which criteria were they chosen?
The definition of core OTUs is not clear, preventing a good understanding of this section. The type of correlation used in networks should be specified. In addition it is not clear why, in some modules (e.g. ‘blue’), 2 networks are shown (Fig 7 a).
It would be interesting to identify specific OTUs characteristic of each type of sample (i.e. divided according to the water cover). However, the way the authors present it (L290-299) lacks of statistical support and therefore is not conclusive. Especially, the authors should take into account the poor triplicate reproducibility in some times (e.g. Sed1_3, Table S5). The links between sampling sites and geochemical properties is absolutely not clear (L299-302).
The comparisons of physicochemical parameters between sites is not supported by statistical tests.
Discussion:
The phrasing must be deeply revised. The claim to study the effect of water table level does not correspond to any significant result. Correlations with environmental variables at the phylum level are meaningless due to the large phenotypic variability within a phylum. Regarding the influence of anthropogenic activity: is there any evidence of connectivity between the Great Wall station and Kitiesh lake supporting the authors suggestion? According to the site topography it seems highly improbable. Can the authors elaborate and discuss on why phosphate would be a driver for Bacteria while nitrite would be a driver for Archaea? Or can it be an artefact? More generally, can the authors elaborate and discuss on the physiology of certain taxa possibly explaining the observed correlations with physicochemical parameters and/or predominance in certain sites?
Minor comments:
L91: what does “warm” mean (quantitatively) in the Antarctica? PCR primers are usually given forward first, reverse second
L139-144: It is unclear if this paragraph is talking about sequence treatment through bioinformatic analysis. If yes, then this should be after the sequencing step description.
L160: representative of what?
L170: which correlation metrics?
Table 1: why giving all replicate values if the average and standard deviation are sufficient. Should specify if the letters ‘a’ and ‘b’ are per environmental parameter. In that case, why do Si and phosphate begin with the letter ‘b’?
L187-195: avoid terms like “best”, “good” for description of results
L291-292: There is no such Tables S6 and S7 available. More generally there is confusion in the orderof supplementary tables throughout the manuscript.
Naming modules on the basis of their color is not rigorous especially for color-blinded readers.
Author Response
Thanks for your constrctive comment and our responses have been uploded as an attachment.
We hope that the revisions are acceptable, and we look forward to hearing from you soon.

Round 2
Reviewer 3 Report (New Reviewer)
see attached file

Author Response
Thanks for your constructive comments. The details can be seen in the attachment. Hope that the revision is acceptable, and we look forward to hearing from you soon.

This manuscript is a resubmission of an earlier submission. The following is a list of the peer review reports and author responses from that submission.
Round 1
Reviewer 1 Report
Dear authors, I have read the present work with attention.
I find that the layout of the text is correct. I wonder about the title, because already in the abstract the authors mention climate change, perhaps it is worth noting this also in the title, necessarily in the words - key words. This will perhaps create more interest in this text for a wider range of readers.
Perhaps for Figures 2 and 3 use a logarithmic scale? For small values in particular, it could be more readable....
These are just my suggestions, as I rate the work highly. It is written in a clear and reliable manner.
I wonder about the literature items, I think there could be more.
Response to Reviewer 1 Comments
Point 1: I wonder about the title, because already in the abstract the authors mention climate change, perhaps it is worth noting this also in the title, necessarily in the words - key words. This will perhaps create more interest in this text for a wider range of readers.
Response 1: We have put “climate change” into keywords.
Point 2: Perhaps for Figures 2 and 3 use a logarithmic scale? For small values in particular, it could be more readable.
Response 2: Thank you for your advice, but the purpose of Figures 2 and 3 in this paper is to find phyla and genera with higher abundance. Our focus is not on the lower ones. If you need specific abundance data, we can also provide it. Thanks again for your suggestion. We will consider using the logarithmic scale in similar situations in future research.
Reviewer 2 Report
General comments:
The manuscript entitled “Diversity analysis of bacteria and archaea in soils of the Kitezh Lake area, Antarctica” is an interesting manuscript which focusing bacterial and archaeal community in Antarctic seasonal meltwater lake. The beginning of introduction sets scene very nicely but the authors did not clearly review related research and the hypotheses were not provided. Please see comments below for several ideas that need justification and/or more details. Some parts of the approaches need more clarification (see comments below for details). Figures in results need major revision. Discussion and conclusion will need to be revised when the concerns have been addressed. Overall, I think that the manuscript idea is interesting and will be valuable for the scientific community. However, additional work is needed to address concerns and improve several parts of this manuscript.
Title
The title indicates what this research is about but it does not provide any information about key results. I would suggest revising the title to be more specific. What is the key result?
Simple Summary
Since this research focuses only on bacteria and archaea, I would suggest changing “microbial” to “bacterial and archaeal”.
Abstract
Line 29: I would suggest removing “widely believed”.
In general, the abstract can be improved by being more specific. I think this would be better when the authors tried to address the hypotheses which are not presented. For example, Line 39 - 41, what does “play important roles” means in this case?
Introduction
Line 53 – 54: What does “amplify the air warming” mean?
Line 56 – 57: “Some studies have reported that microorganisms are very sensitive to environmental changes [9-12]” – What did they find? These are four studies? Please clearly explain how these studies support this statement.
In general, this paper focused on bacterial community, but very little information about bacterial community in Antarctica was introduced. There are other studies which also focused on polar bacterial community, please introduce them here. What have they done? What were the general polar bacterial composition? Were there any patterns that other researchers found about bacterial community? Is there a gap in bacterial research that this paper will address and help moving the field forward? Why below-peak, intertidal, and sediment soils would be great to study? Lastly but MOST IMPORTANTLY!, What are the hypotheses of this research? (Major concern #1)
Materials and Methods
Line 95: How many samples were collected (in total)? – There are 9 sample? Major concern#2: Is this sample collection sufficient to answer the research question?
Major concern#3: many software and packages were used in the analysis but were not cited. This is not appropriate.
Figure 1: SED is sediment. However, the location shown on the picture doesn’t seem to be under water as described in Simple Summary (Line 19:SED (always under water)). Please justify.
Results
Line 188: Instead of using Table S1, alpha diversity plot with statistical analysis would be much appropriate.
Line 209: there seems to be extra space between “….dominance.” And “And at the genus…”
Line 209: “41 known genera” may be use “identifiable genera” instead
Figure 2 and 3: labels can’t be read.
Figure 6 can’t be read
Figures need major revison.
Discussion and Conclusion
Without hypotheses, the discussion doesn’t seem to be meaningful. Please add hypotheses to the paper and frame the discussion based on the hypotheses.
Response to Reviewer 2 Comments
Point 1: The title indicates what this research is about but it does not provide any information about key results. I would suggest revising the title to be more specific. What is the key result?
Response 1: The title has been changed to “Soil geochemical properties influence diversity of bacteria and archaea in soils of the Kitezh Lake area, Antarctica”.
Point 2: Since this research focuses only on bacteria and archaea, I would suggest changing “microbial” to “bacterial and archaeal”.
Response 2: The “microbial” has been changed to “bacterial and archaeal”.
Point 3: Line 29: I would suggest removing “widely believed”.
Response 3: It has been deleted.
Point 4: In general, the abstract can be improved by being more specific. I think this would be better when the authors tried to address the hypotheses which are not presented. For example, Line 39 - 41, what does “play important roles” means in this case?
Response 4: The abstract has been reorganized and the specific hypothesis is presented. As for the meaning of“play important roles”, we did LEfSe at the phylum classification to explain (LDA>4). Take bacteria as an example, it can be seen from the results that the BPK has the highest pH value and the INT has the lowest one. The biomarkers of BPK are Firmicutes and Proteobacteria, while the biomarkers of INT are Actinobacteria, Bacteroidetes and Acidobacteria. So we supposed that the geochemical properties between different sampling sites may lead to differences in bacterial and archaeal community structure by changing the abundances of some key bacteria and archaea.
Point 5: Line 53 – 54: What does “amplify the air warming” mean?
Response 5: The meaning of “amplify the air warming” in the literature have been specifically described as “As temperature rises up, lakes open earlier and more soils would absorb more solar energy in summer. Those heat could be transferred to water column in winter, and ice cover would reduce heat loss. Both making water warming be amplified than air warming.”
Point 6: Line 56 – 57: “Some studies have reported that microorganisms are very sensitive to environmental changes [9-12]” – What did they find? These are four studies? Please clearly explain how these studies support this statement.
Response 6: References that were not sufficiently relevant were removed. The original text has been changed to “According to Vincent [9], the Arctic microbiota were sensitive to exhibit rapid changes under global warming and can be viewed both as sentinels and amplifiers of global change. In the Antarctic region, bacterial communities have been shown to adapt more quickly to temperature rising at higher temperatures than at lower temperatures[10]. Microbial communities can be structured by the interactions between geochemical conditions and microbial capabilities mechanistically[11].”
Point 7: In general, this paper focused on bacterial community, but very little information about bacterial community in Antarctica was introduced. There are other studies which also focused on polar bacterial community, please introduce them here. What have they done? What were the general polar bacterial composition? Were there any patterns that other researchers found about bacterial community? Is there a gap in bacterial research that this paper will address and help moving the field forward? Why below-peak, intertidal, and sediment soils would be great to study? Lastly but MOST IMPORTANTLY!, What are the hypotheses of this research? (Major concern #1)
Response 7: The introduction has been revised. The description of the results of the analysis of microbial diversity in the Antarctic region has been added, the purpose of the setting of below-peak, intertidal, and sediment soils has been explained, and the hypothesis of this study has been added.
Point 8: Line 95: How many samples were collected (in total)? – There are 9 sample? Major concern#2: Is this sample collection sufficient to answer the research question?
Response 8: Yes, there are 9 samples from 3 sampling sites. We believe that the sample size is sufficient, because the three types of soil are well characterized, and we set three parallel samples for each sampling site.
Point 9: Major concern#3: many software and packages were used in the analysis but were not cited. This is not appropriate.
Response 9: References to methods and software used have been added.
Point 10: Figure 1: SED is sediment. However, the location shown on the picture doesn’t seem to be under water as described in Simple Summary (Line 19:SED (always under water)). Please justify.
Response 10: The sampling sites marked in the Figure 1 is not accurate and has been corrected.
Point 11: Line 188: Instead of using Table S1, alpha diversity plot with statistical analysis would be much appropriate.
Response 11: The plots has been added as Figure S1a and Figure S1b.
Point 12: Line 209: there seems to be extra space between “….dominance.” And “And at the genus…”
Response 12: The extra space has been removed.
Point 13: Line 209: “41 known genera” may be use “identifiable genera” instead
Response 13: The text “known” has been replaced with “identifiable” or “classifiable”.
Point 14: Figure 2 and 3: labels can’t be read. Figure 6 can’t be read. Figures need major revison.
Response 14: The font size in the figures has been adjusted.
Point 15: Without hypotheses, the discussion doesn’t seem to be meaningful. Please add hypotheses to the paper and frame the discussion based on the hypotheses.
Response 15: The indication of the hypothesis has been reorganized and some relevant content has been added to the discussion section.
Reviewer 3 Report
Dear authors, please find my review in the attached document.

Response to Reviewer 3 Comments
Point 1: Authors state in lines 31-32 (abstract) that their study aimed to investigate the effects of climate warming on soil microbial communities. However, the only relationship between their work and climate is the fact that soil samples were collected in a meltwater seasonal lake shore. This is a weak relationship. There are no data about local parameters, for example snow coverage, precipitation, rate of glacial melting, average air temperature, etc. Despite the logical relationship between Antarctic climate change and seasonal lakes, authors should provide climate data for the studied area. Otherwise, it is not possible to assume that microbial ecological data is a result of climate change or natural seasonal variations.
Response 1: The Kitezh Lake is located approximately 3.4km southwest of Collins Glacier and is virtually unaffected by glaciers. The sampling period is 20 days in summer, the sampling area was free of snow and ice, and the air temperature fluctuates between -3-8℃. The temperature when sampling is 2℃. The average annual temperature on the Fildes Peninsula is -2.1℃, with average summer temperatures above 0℃for four months, and rainfall between 350 and 500 mm per year.
Point 2: Introduction has a clear focus on methodology, not the biological phenomena. I suggest improving the introduction by describing the Antarctic soil microbial diversity, the main soil groups found, etc. At the end, authors could write about the methods other people used to study Antarctic soil diversity, including the lack of works using WGCNA.
Response 2: The following has been added to the introduction:
In the study of microbial community diversity, from the study of Chong, et al, the distribution patterns of soil bacteria in different regions of Antarctica were summa-rized[16]. At a higher taxonomic level, such as phylum and class, the soil bacterial community composition can be highly stable, while at lower taxonomic levels, it might be sensitive to both spatial and environmental gradients. Bacteroides were the most widespread phyla(89%) of the study sites, followed by actinomycetes (86%) and acid bacteria (77%). Regional community differences may be caused by latitude, climate and geological characteristics. In Sediments of West Lake Bonney, McMurdo Dry Valleys, Bacteriodetes and the Proteobacteria were found to be major bacterial phyla, while Thaumarchaeota and the Crenarchaeota were the main archaea[17]. Firmicutes and Deltaproteobacteria and Epsilonproteobacteria as dominant bacteria, and Thaumar-chaeota as major archaea were identified in cryoconite ecosystems[18]. The dominant phyla in soil samples from the Keller Peninsula, King George Island, Antarctica were Bacteroidetes, Acidobacteria, Betaproteobacteria, Alphaproteobacteria, and Actinobac-teria[19]. Given that the structure of microbial communities varies from place to place, in our research, the Kitezh Lake, Fildes Peninsula, King George Island, Antarctica was chosen to be the research area to study the diversity and composition of bacterial and archaeal communities in soils.
Point 3: It is not clear why different multivariate analysis were used for Bacteria (CCA) and Archaea (RDA). Check line 161 and please provide some explanation in the manuscript.
Response 3: The method of selecting RDA/CCA has been specifically described as “To find key geochemical factor affecting community composition of bacteria and ar-chaea in soil, Detrended Correspondence Analysis (DCA), CCA and RDA were per-formed in R. Through the results of DCA, when the axis length of DCA1 is smaller than 3, RDA is better than CCA, when it is bigger than 4, CCA is preferred, and when it is between 3 and 4, both RDA and CCA can be selected.”
Point 4: There are several mistakes when reporting Tables, especially the supplementary tables. For example, in line 197 the alpha index of archaea is said to be in Table S2 according to the manuscript. However, it is in the second tab of Table S1. The same in line 225, where Table S3 should be Table S2. Also in line 258, where Table S5 should be Table S4. Please check all tables if they are reported correctly.
Response 4: Sorry for the trouble. All tables have been checked.
Point 5: Some results are not discussed appropriately. For example, in Figure 2 it is shown INT with higher diversity than other samples. Why this happened at Phylum and genus levels? How is the INT environment related to this result? Does the water tide cause a more dynamic environment?
Response 5: The following text has been added in the discussion section: “We hypothesized that the higher diversity of bacterial communities in the INT environment was mainly due to the greater diversity of environmental factors at this sampling site. Firstly, compared with BPK (below-peak soils), INT (intertidal soils) has access to water, which is richer in nutrients than the soil. So it is easier to gain nutrients from water, which is more conducive to the growth of microorganisms. In addition, compared with SED (sediments), INT is exposed to sufficient oxygen, which is more conducive to the growth of aerobic bacteria. Taking these two factors together, INT may have higher microbial community diversity. Related statements have been added to the discussion.”
Point 6: In lines 340-342 authors claim that Firmicutes were positively correlated with pH and present the references of Khamami et al. (2021) and Zhao et al. (2019) as base of comparison. However, there are two important observations here. First, authors used Spearman to correlate positively the Firmicutes abundance in BPK with pH. Since pH is a non-linear variable (e.g. gaussian distribution), the Spearman correlation analysis is not the best choice. A positive relationship means that increasing Firmicutes abundance is correlated with increasing pH (alkaline). A negative relationship means the opposite, a correlation to decreasing pH (acid). Therefore a negative correlation does not mean lack of relationship, but a correlation with acid pH as expected and seen in Table 4 for Acidobacteria (r = -0,917). So in Line 341 the sentence should state that Firmicutes is correlated with alkaline pH.
Secondly, the references of Khamami et al. (2021) and Zhao et al. (2019) are not suitable for comparison, since they found a relationship between Firmicutes and (alkaline) pH in different environments. Khamami worked with anaerobic waste reactors, and Zhao worked with copper mining soils. Please use references that found Firmicutes in alkaline soils of Antarctica.
Response 6: Firstly, for the selection of Spearman, we analyzed the distribution of bacteria and the difference of geochemical properties among sampling sites. Because the two did not conform to the normal distribution, we chose spearman instead of pearson. And we cited an referance (doi:10.3390/microorganisms8081202.) in the method section. Secondly, what we want to express the meaning of positive correlation and negative correlation is also as you described. However, the expression may get wrong, thank you so much for your advice. Finally, the unsuitable references has been replaced.
Point 7: Line 74: Write Carini in capital letters.
Response 7: It has been corrected.
Point 8: Line 79: remove the word “believed”
Response 8: The word has been removed.
Point 9: Line 95: Describe how much soil (in grams or kilograms) was collected in each site.
Response 9: It has been specifically described in the method as “Fifty grams of topsoil (0-5 cm) was collected into TWIRL’EM sterile bags (Labplas Inc., Sainte-Julie, QC, Canada) using a sterile shovel for each sample.”.
Point 10: Line 107: Use “Ten grams of soil”. Avoid using numbers to start a sentence.
Response 10: The text has been changed.
Point 11: Line 108: If Antarctic grasses are present in the soil sites, please describe which species (Deschampsia antarctica, Colobanthus quitensis or both).
Response 11: There are little Deschampsia antarctica in this area, none in the sampling area.
Point 12: Line 122-123: Primer sequences do not correspond to the described in the original papers. Please check Caporaso et al. 2011 and Muyzer et al. 1993 for primers 806R and 341F, respectively.
Response 12: In this study, the universal primers in the V3-V4 region of bacteria were used, which were verified to be correct, and the primers in the literature were specific for different bacterial species.
Point 13: Line 137: Change “bright strip” for “amplification signal” or “amplification band”. Also, please explain if all samples presented an unique band/signal.
Response 13: The phrase has been changed. All samples presented an unique signal.
Point 14: Line 147: Please remove “which is an ideal next generation sequencing platform” because it is a subjective statement. Illumina MiSeq may be appropriate for some studies, but not for other.
Response 14: The sentence has been deleted.
Point 15: Line 164: Please insert a citation of the WGCNA package. Also, start the sentence with “The R package…” instead of “An R package…”
Response 15: The text has been changed and a citation have been inserted(doi:10.1186/1471-2105-9-559).
Point 16: Line 217: Please use formal writing, remove “we can see that”.
Response 16: The text has been removed.
Point 17: Line 255: Results describe 10 phyla and 18 genera. However Table S2 show a total of 16 phyla and 20 genera. Is this correct? Please check
Response 17: The content in the supplementary tables has been corrected.
Point 18: Line 268-269: Avoid starting a sentence with “And”. My suggestion is: “A total of 3 and 1 genera/genus…”
Response 18: The sentence has been modified.
Point 19: Line 271: Table S4 should be Table S2, correct?
Response 19: Yes, sorry for the trouble. All supplementary tables has been corrected.
Point 20: Line 277: Table 4 is too big, consider moving it to supplementary materials.
Response 20: Table 4 has been removed and the content can be seen in Table S3.
Point 21: Line 331: Please use formal writing. Change “It´s” for “It is”
Response 21: Similar expressions in the manuscript have been changed.
Point 22: Line 353-355: Not all Crenarchaeota are ammonia-oxidizing, so I suggest to not make generalizations at Phylum level.
Response 22: The description of Crenarchaeota has been changed to “Crenarchaeota is often found to be major archaea as anaerobic, thermophilic and aci-dophilic phylum[51] which comprises the widely existent ammonium oxidized archaea”.
Point 23: Figure 2. Please increase font size.
Figure 3. Please increase font size.
Figure 4. Please increase font size.
Figure 5. Please increase font size.
Figure 6. Please increase font size
Response 23: Font size in figures has been adjusted.